# Salivary Antibody Responses to Two COVID-19 Vaccines following Different Vaccination Regimens

**DOI:** 10.3390/vaccines11040744

**Published:** 2023-03-28

**Authors:** Hassan Alkharaan, Hatem Al-Qarni, Muath A. Aldosari, Mohammed Alsaloum, Ghada Aldakheel, Mohammed W. Alenazi, Naif Khalaf Alharbi

**Affiliations:** 1Department of Preventive Dental Sciences, College of Dentistry, Prince Sattam Bin Abdulaziz University, Al-Kharj 16278, Saudi Arabia; 2Department of Restorative and Prosthetic Dental Sciences, College of Dentistry, King Saud Bin Abdulaziz University for Health Sciences, Riyadh 14611, Saudi Arabia; 3King Abdullah International Medical Research Center, Ministry of National Guard Health Affairs, Riyadh 14611, Saudi Arabia; 4Department of Oral Health Policy and Epidemiology, Harvard School of Dental Medicine, Boston, MA 02115, USA; 5Department of Periodontics and Community Dentistry, King Saud University College of Dentistry, Riyadh 12372, Saudi Arabia; 6Vaccine Development Unit, King Abdullah International Medical Research Center (KAIMRC), Riyadh 11481, Saudi Arabia; 7King Saud Bin Abdulaziz University for Health Sciences, Riyadh 14611, Saudi Arabia

**Keywords:** COVID-19, SARS-CoV-2, vaccines, saliva, IgG, mucosal, antibody, immune responses, ELISA

## Abstract

***Background:*** To date, little is known about the salivary mucosal immune response following different COVID-19 vaccine types or after a booster (3rd) dose of the BNT162b2 (BNT) vaccine. ***Methods:*** A total of 301 saliva samples were collected from vaccinated individuals and arranged into two cohorts: cohort 1 (*n* = 145), samples from individuals who had received two doses against SARS-CoV-2; cohort 2 (*n* = 156), samples from individuals who had received a booster of BNT vaccine. Cohorts 1 and 2 were sub-stratified into three groups based on the types of first and second doses (homologous BNT/BNT, homologous ChAdOx1/ChAdOx1, or heterologous BNT/ChAdOx1vaccinations). Salivary immunoglobulin G (IgG) response to SARS-CoV-2 spike glycoprotein was measured by ELISA, and clinical demographic data were collected from hospital records or questionnaires. ***Results:*** Salivary IgG antibody responses against different vaccines, whether homologous or heterogeneous vaccination regimens, showed similar levels in cohorts 1 and 2. Compiling all groups in cohort 1 and 2 showed significant, albeit weak, negative correlations between salivary IgG levels and time (r = −0.2, *p* = 0.03; r = −0.27, *p* = 0.003, respectively). In cohort 2, the durability of salivary IgG after a booster dose of BNT162b2 significantly dropped after 3 months compared to the <1 month and 1–3 months groups. ***Conclusions:*** Different COVID-19 vaccine types and regimens elicit similar salivary anti-SARS-CoV-2 IgG with modest waning over time. Boosting with BNT162b2 vaccine did not produce an evident increase in mucosal IgG response whereby COVID-19 recovered subjects show higher salivary IgG than naive, post-vaccination subjects. The ChAdOx1/ChAdOx1 regimen showed better correlation between salivary IgG levels and durability. These findings highlight the importance of developing oral or intra-nasal vaccines to induce stronger mucosal immunity.

## 1. Introduction

In late 2019, the highly contagious severe acute respiratory syndrome coronavirus 2 (SARS-CoV-2) was first identified in Wuhan, China, and spread across the globe, causing the coronavirus disease 2019 (COVID-19) pandemic [1]. As of January, 2023, the reported cases of COVID-19 reached over 661 million with 6.6 million mortalities [2]. The COVID-19 pandemic presented an unprecedented public health crisis that led to a remarkable global collaboration for rapid vaccine development. Among authorized vaccines, the mRNA-based vaccine (BNT162b2, from Pfizer–BioNTech) and adenoviral ChAdOx1-based vaccine (AZ1222, from AstraZeneca, Cambridge, UK), have both shown significant efficacy to reduce severe symptoms of COVID-19 and limit mortality [3,4,5,6]. These two vaccines are approved in two doses, prime and boost vaccination regimens, while a third dose of BNT162b2, also known as the booster dose, has been approved to maintain long-term protection against the viral infection [7].

The course of the immune responses induced by the vaccination has been shown to correlate with several factors, including age, gender, comorbidity, previous SARS-CoV-2 infections, duration between prime and boost vaccine doses, and number of boost doses as well as the circulating virus variants [8,9,10,11,12,13]. Likewise, vaccination regimens with heterologous or homologous vaccine platforms have been shown to elicit a range of immune responses in serum, with superiority of heterologous ChAdOx1-mRNA or homologous mRNA-mRNA dosing regimens over homologous ChAdOx1-ChAdOx1 vaccination [14,15].

Systemic humoral and cellular immunity mediated by memory B- and T-cells, respectively, are rapidly induced by booster doses after viral infection or vaccination [14,16,17,18], but whether the current intramuscular COVID-19 vaccines can boost a similar mucosal immune response after the third dose is still not well-characterized.

Mucosal immunity in the upper respiratory tract is considered a front-line defense against SARS-CoV-2 invasion [19,20]. The oral cavity is another important site for SARS-CoV-2 infection, and fractions of cellular or acellular saliva may transmit SARS-CoV-2 infection [20]. It is also an accessible mucosal site for sampling. SARS-CoV-2 principal entry factors, such as the ACE2 and TMPRSS2 cellular receptors are expressed in the oral tissues (buccal mucosa, ventral tongue, and the dorsal tongue). Mucosa of the nasal cavity, oropharynx, and oral epithelial cells are also shed in saliva [20,21].

Saliva is an efficient diagnostic biofluid to monitor mucosal antibody response against SARS-CoV-2, even in mild cases [22,23]. Antibody levels generated by B-cells into circulation and mucosal sites have been reported to negatively correlate with the risk of SARS-CoV-2 infection [22,24,25,26]. Salivary and serum specific IgG antibody responses have been found to be strongly boosted and longer lasting after COVID-19 systemic vaccine while specific IgA antibody responses were minimal, short-lived, and impacted by pre-exposure to SARS-CoV-2, yet associated with protection against breakthrough infection [22,26,27,28]. However, mucosal antibody responses in saliva against SARS-CoV-2 following a booster (3rd dose) of heterologous or homologous dosing strategy are still unclear.

In this study, we measured spike-specific anti-SARS-CoV-2 IgG levels in saliva samples using an in-house ELISA that has previously been validated for seroprevalence studies [29], following 2nd and booster doses from participants who had received either heterologous or homologous vaccination of ChAdOx1 and BNT162bn2 (BNT) vaccines. We also investigated whether levels of vaccine-induced anti-spike IgG differed in recovered COVID-19 individuals.

## 2. Materials and Methods

### 2.1. Study Design and Sample Collection

The study was approved by the research ethics committee at Prince Sattam Bin Abdulaziz University (REC-HSD-114-2022) and complied with the Declaration of Helsinki. All study subjects signed an informed consent form prior to study participation at the dental clinics of Prince Sattam Bin Abdulaziz University and King Saud Bin Abdulaziz University for Health Sciences Riyadh province, Saudi Arabia, during November 2021 to April 2022. A total of 301 saliva samples were collected from vaccinated individuals and arranged into two cohorts: cohort 1 (*n* = 145), samples from individuals who had received two doses against SARS-CoV-2; cohort 2 (*n* = 156), samples from individuals who had received a booster (3rd dose) of mRNA-based vaccine. Cohorts 1 and 2 were sub-stratified into three groups based on the types of first and second doses (homologous BNT/BNT, homologous ChAdOx1/ChAdOx1, or heterologous BNT/ChAdOx1 vaccinations). Cohort 2 was grouped according to the time since booster dose into three subsets: (1) less than 1 month; (2) 1–3 months; and (3) more than 3 months after the booster dose. Clinical demographic data were collected from hospital records or questionnaires, and previous exposure to COVID-19, vaccination date, and type information were compiled from a governmental record application (Tawakkalna) approved by the Saudi Ministry of Health. The COVID-19 diagnosis was confirmed in all convalescent patients using the SARS-CoV-2 reverse transcription polymerase chain reaction (RT-PCR). All participants with immunosuppressive diseases and/or therapy, pregnancy, or autoimmune disorders were excluded from the study.

Unstimulated saliva was collected under the supervision of dental professionals before dental treatment. All participants were instructed to refrain from eating, drinking, smoking, or teeth brushing at least 1 h prior to passively drooling saliva into sterile 2 mL Eppendorf tubes. All samples were stored at −80 °C within 12 h.

### 2.2. SARS-CoV-2 Antibody Detection by ELISA

An in-house enzyme-linked immunosorbent assay (ELISA) was used to detect IgG antibodies against SARS-CoV-2 in saliva samples. The ELISA was performed following the previously reported procedure in [30]. Briefly, Nunc MaxiSorp 96-well ELISA microplates (Thermo Fisher, Waltham, MA, USA) were coated with a recombinant S1 subunit of the SARS-CoV-2 spike protein (Sinobiological Ltd., Beijing, China (Cat. No. 40591-v08B1) at a concentration of 1 μg/mL. The plates were incubated at room temperature (RT) overnight. The next day, the plates were washed six times with phosphate buffered saline (PBS) with 0.5% Tween20 (PBS-T) using automated microplate washer (Molecular Devices, San Jose, CA, USA). The plate wells were blocked by 100 μL washing buffer containing 10% skimmed milk (blocking buffer) for 1 h at RT. The saliva samples were diluted 1:100 in PBS-T, and 50 μL of each diluted sample were added into duplicate wells and then incubated for 2 h at RT. Then, 50 μL of 1:1000 diluted alkaline phosphatase, labeled goat anti-human IgG secondary antibody (Thermo Fisher, Waltham, MA, USA), were added and incubated for 1 h at RT. The plates were then washed six times, and the pnitrophenylphosphate (PNPP) substrate dissolved in diethanolamine buffer and deionized water was added. The optical density (OD) at 405 was measured using a microplate reader (Molecular Devices, San Jose, CA, USA). The cut-off value was set as the average of the negative control serum samples plus three times the standard deviation. The negative control samples were sera collected before the COVID-19 pandemic, and the positive control serum samples were from confirmed recovered COVID-19 cases. The same control samples were aliquoted and used in every ELISA run to limit freeze-thaw effects.

### 2.3. Commercial Kit Validation

Performance of the in-house assay was validated by a commercial ELISA kit, the anti-SARS-CoV-2 IgG ELISA kit (Euroimmun, Schleswig-Holstein, Germany), according to the manufacturer’s instructions. Readouts were reported as the ratio of sample optometric density (OD) over the OD of an internal calibrator. Positive and negative controls provided with the kit were always included in each run.

### 2.4. Statistical Analysis

All statistical analyses were performed using GraphPad Prism Version 9.0. Pairwise analysis between each group was calculated using the Kruskal–Wallis test with Dunn’s multiple comparisons correction for quantitative data and Fisher’s exact test for qualitative data. Correlation analysis between anti-spike (developed in-house) IgG and anti-spike (commercial kit) IgG, and between anti-spike (developed in-house) IgG and time, was conducted using a Spearman correlation analysis.

## 3. Results

### 3.1. Characteristics of the Study Participants

A total of 301 vaccinated participants were recruited for this study and arranged into two cohorts based on the number of doses received. In cohort 1 (*n* = 145), saliva samples from individuals who received two doses were grouped based on the vaccine types into three groups: homologous BNT/BNT (*n* = 65), homologous ChAdOx1/ChAdOx1 (*n* = 38), and heterologous BNT/ChAdOx1 (*n* = 42). The homologous ChAdOx1/ChAdOx1 group had fewer female and recovered COVID-19 individuals than the BNT/BNT group (Table 1). Cohort 2 (*n* = 156) comprised the saliva samples from participants who received a booster dose of BNT vaccine; the median age was similar in all of the groups, but the age range of the ChAdOx1/ChAdOx1/BNT group was smaller than that of the BNT/ChAdOx1/BNT group (Table 2).

### 3.2. Validation of the In-House ELISA

To examine the validity of the salivary IgG readouts in this study, a commercial ELISA kit was compared to the in-house ELISA using a panel of 21 saliva samples (Figure 1A). This assay presented a significant positive correlation between the in-house and commercial ELISA (r = 0.6, *p* = 0.007), indicating reliable results of the in-house ELISA used in this study to the commercial ELISA kit.

### 3.3. Different Vaccine Types and Regimens Elicit Similar Salivary Anti-SARS-CoV-2 IgG

To evaluate the mucosal immune response elicited by different types of COVID-19 vaccines, anti-spike IgG was measured in the saliva of non-infected COVID-19 individuals based on their received vaccine types. Interestingly, the levels of salivary IgG induced upon different vaccinations, whether homologous or heterogeneous vaccination regimens, showed similar results in cohorts 1 and 2 (Figure 1B,C). This suggests a similar stimulation of salivary IgG by mRNA-based and adenovirus-based vaccines against SARS-CoV-2.

### 3.4. Waning of Salivary IgG Levels after BNT162b2 Booster

To evaluate whether the intramuscular BNT162b2 booster dose induces stable mucosal immunity, the durability of anti-spike IgG in the saliva of non-infected participants was examined according to the time since the booster dose, combining all primary vaccination regimens. Salivary IgG levels significantly dropped after 3 months compared to the <1 month and 1–3 months groups (*p* = 0.002; *p* = 0.003, respectively), as shown in Figure 1D. Next, salivary anti-spike IgG levels before and after the BNT booster dose (cohort1 versus cohort 2, respectively) were compared (Figure 1E). The salivary IgG levels were relatively similar in the two cohorts; however, the time intervals from the last dose to sample collection was shorter in cohort 2 compared to cohort 1 (Appendix A). These findings of reduced longevity indicate a moderate activation of mucosal immunity by systemic booster vaccination.

### 3.5. Impact of Previous COVID-19 Infection on the Levels of Vaccine-Induced Salivary IgG

The impact of previous SARS-CoV-2 infection on mucosal immune responses was evaluated. Salivary IgG levels in previously COVID-19 infected individuals were higher than in naive vaccinated individuals (Figure 1F). Moreover, to investigate the influence of all potential risk factors (gender, age, BMI, smoking, previous COVID-19 infection, and days since sampling) on salivary IgG levels, multiple linear regression was performed and revealed that previous COVID-19 infection is the only contributing factor to IgG levels in saliva (Table 3). 

### 3.6. Persistence Patterns of Salivary IgG Levels following Different Vaccination Regimens

The persistence pattern of salivary IgG over time was shown to wane over 3 months, as shown above. In order to further characterize it according to vaccination strategies and vaccine type, only the time since the last dose was analysed in relation to the salivary IgG levels. Samples of previously infected COVID-19 individuals were excluded. Both cohorts showed modest waning of salivary IgG over time post-last dose, regardless of the received vaccine, except for the groups who received first and second doses of homologous ChAdOx1/ChAdOx1, in which IgG levels were relatively stable over time (Figure 2A,B). This indicates efficient longevity of immune responses induced by viral vectored vaccines. Interestingly, the waning pattern of homologous BNT/BNT/BNT in cohort 2 was faster as compared to the other groups (r = –0.32, *p* = 0.007). Collectively, compiling all groups of vaccination regimens and types in each cohort showed significant, albeit weak, negative correlations between salivary IgG levels and time (r = –0.2, *p* = 0.03; r = –0.27, *p* = 0.003, respectively) (Appendix A).

## 4. Discussion

Despite the effective control of the COVID-19 pandemic by vaccination, there are still some concerns for healthcare systems and authorities, including high rates of breakthrough infections, emergence of new virus variants, waning of vaccine-induced immunity over time, and the unknown long-term potential risks of vaccine adverse effects [31,32,33]. While most studies have focused on systemic immune responses conferred by COVID-19 vaccines, this study focused on mucosal immunity in saliva following different types and regimens of vaccines. Here, we reported that COVID-19 vaccination of homologous or heterologous vaccine types induce similar salivary anti-SARS-CoV-2 IgG levels, and participants who received the primary vaccination dosage (two doses) have similar salivary IgG as compared to those who received a booster dose (three doses).

We showed that hybrid immunity of previous infection and vaccination elicits greater salivary IgG levels than immunity conferred by only vaccination in gender and age-matched cohorts. The natural infection induces immune responses towards various antigens and epitopes of many of the viral proteins, unlike the vaccine-specific immunogenicity that is focused only on the spike antigen. The route of the natural infection is primarily mucosal, which could have contributed to the high level of salivary IgG responses. Taken together, these suggest that systemic vaccination could still boost the mucosal immune responses and support the efforts of developing mucosal COVID-19 vaccines.

SARS-CoV-2 can utilize the oral mucosae and salivary glands for infection and replication [20]. The infected epithelial cells are also shed in the saliva and can transmit SARS-CoV-2 infection [20]. Therefore, antibody detection in the saliva is a practical diagnostic tool that may deliver information about the susceptibility to SARS-CoV-2 infection and transmission. Several studies have shown that vaccination could induce mucosal antibody responses that might contribute to protection against mucosal infection of respiratory pathogens [20,26,28,34]. Vaccine types and number of doses have been shown to affect antibody responses in serum [14,15,35]. Barros-Martins et al. [14] reported that the heterologous dosing with ChAdOx1/BNT approach resulted in greater humoral IgG and IgA immune responses in serum than homologous ChAdOx1/ChAdOx1. Similar results were also reported by Schmidt et al. [35], showing higher vaccine-induced anti-spike IgG levels in serum after heterologous ChAdOx1/BNT as compared to homologous ChAdOx1/ChAdOx1; however, the participants dosed with heterologous ChAdOx1/BNT showed similar titers of anti-spike IgG antibody in serum against SARS-CoV-2 to those who received homologous BNT/BNT [14]. Recently, Mubarak et al. evaluated salivary IgG against the SARS-CoV-2 spike protein from participants who received two doses of the primary vaccination cycle and reported insignificant differences in IgG titers between homologous and heterologous vaccination strategies [36]. In line with Mubarak et al.’s study, our data further confirm that vaccine-induced anti-spike IgG remains stable in saliva, even after a third dose boosting by BNT, regardless of the primary vaccination strategy.

Our previous work showed that salivary IgG against SARS-CoV-2 persists up to 9 months after recovering from mild COVID-19 infection [22]. We here extend these findings by showing that, among the different analyzed risk factors in our study, exposure to COVID-19 infection is the only factor that showed a significant impact on the levels of salivary anti-spike IgG in vaccinated individuals, suggesting the importance of the induction of local mucosal immune responses, by natural infection, in eliciting greater salivary IgG against SARS-CoV-2, which can be boosted by COVID-19 vaccination. This supports the idea of priming immune responses with mucosal vaccines for respiratory infections such as SARS-CoV-2, which utilize mucosa as entry site. This result is also in line with previous studies that showed that the combination of natural infection and vaccination offers greater durability of immunity [37,38]. Intriguingly, Goel R. R. et al. reported that COVID-19 recovered subjects showed robust boosting after the first vaccine dose but no increase in circulating antibodies or neutralizing titers after the second dose [16]. Our data showed that participants who received booster doses had similar anti-spike salivary IgG levels to those who received only two doses. The limitation here is that we assessed IgG levels in a shorter time range in the boosted cohort (median days since last dose = 77 days) compared to the cohort who received only two doses (median days since last dose = 137 days). However, a recent study estimated that BNT162b2 effectiveness against any SARS-CoV-2 infection reaches its peak in the third week after the first dose and in the first month after the second dose [33], which were covered by the sampling times in our cohorts. Consequently, whether a single or two doses are enough to prime mucosal immunity to protective levels, especially in previously infected people, needs further investigation.

Waning of antibodies has been well documented in several studies in vaccinated persons and in those who have been infected with SARS-CoV-2 [39,40]. However, some studies have reported that hybrid immunity provides greater levels of protective antibodies against infection [38,41]. The results reported here excluded recovered individuals to assess the patterns of salivary IgG levels following different vaccination regimens. They showed relatively stable salivary IgG levels over time with slight waning patterns in those who received homologous BNT/BNT. This finding is of great interest as it sheds light on the different vaccine-induced responses between systemic and mucosal immunity.

The limitation of this observational study was the potential misclassification of recovered persons as the probability of undiagnosed infection is present; however, a control procedure to this potential bias was performed by collecting COVID-19-like symptoms three months prior to sampling, which revealed no significant differences (Appendix A). Another limitation is targeting the spike protein of the original Wuhan-Hu-1 isolate. Although the continuous circulation of the virus and the selective pressure by antibodies or the immune system have been shown to contribute to the development of SARS-CoV-2 mutated strains, which could escape immune surveillance [32], several studies have reported that vaccine-generated antibodies target conserved epitopes of SARS-CoV-2 spike proteins and induce cross-reactive immune responses against emerging variants of SARS-CoV-2 [42,43,44], implying a potentially efficient diagnostic accuracy for original strain anti-spike IgG to evaluate the effectiveness of vaccines, which were all developed based on the original SARS-CoV-2 strain.

These findings support the recent efforts of moving towards next-generation COVID-19 vaccination. These efforts include the development of adenoviral vector-based COVID-19 vaccines, administered orally or intranasally into mice or hamsters, which have shown strong mucosal immunogenicity and improved protection [45,46]. In conclusion, this study addresses questions about the need for the administration of mucosal vaccines to boost mucosal immunity in order to achieve near-sterilizing immunity and to limit both disease transmission and the emergence of new variants of coronaviruses.

## Figures and Tables

**Figure 1 vaccines-11-00744-f001:**
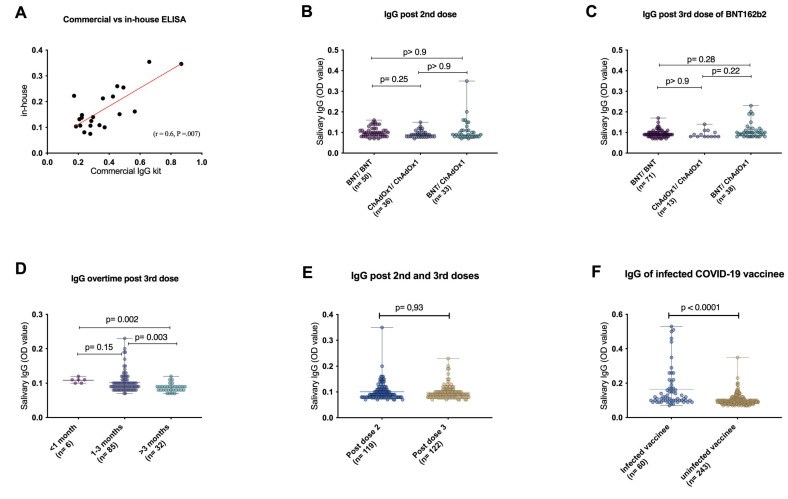
Comparisons between a commercial and the in-house ELISA were performed using a panel of 21 saliva samples (**A**) to validate anti-SARS-CoV-2 spike protein IgG in-house assay performance in saliva. The vaccine-induced antibodies against SARS-CoV-2 Spike-protien between different vaccine types after 2nd dose (**B**) and 3rd dose (**C**) were evaluated in non-infected COVID-19 individuals. Salivary IgG levelsafter BNT booster (cohort 2) were grouped based on time since the 3rd dose into (1) less than 1 month, (2) 1–3 months, and (3) more than 3 months (**D**). The levels of salivary IgG before and after the BNT booster dose (**E**) and between previously COVID-19 infected individuals and non-infected individuals (**F**) were compared. A nonparametric Mann–Whitney U test for significance was performed. BNT, BNT162b2 vaccine. ChAdOx1, ChAdOx1 nCoV-19 vaccine.

**Figure 2 vaccines-11-00744-f002:**
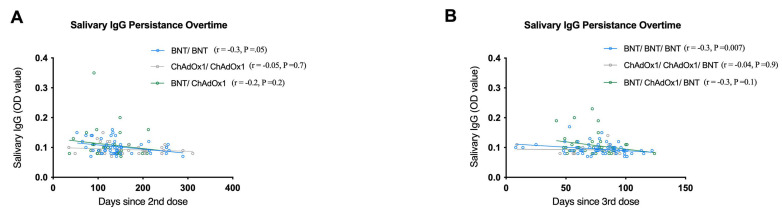
Correlation between salivary IgG levels and time since the 2nd dose (**A**) and 3rd dose (**B**) following different vaccination regimens. Spearman correlation analysis was used to determine rho and *p* values.

**Table 1 vaccines-11-00744-t001:** Demographics of Cohort 1 Study Participants.

		Cohort 1 (*n* = 145)		
		BNT/BNT (*n* = 65)	ChAd/ChAd (*n* = 38)	BNT/ChAd (*n* = 42)
Parameters				
Gender (F:M)		46:19	**17:21 *** **Ω**	22:19
Age (years) median (range)		23 (15–54)	28 (16–52)	25 (14–55)
BMI (kg/m^2^) median (range)		24 (16.8–37)	25 (15.6–42)	23 (14–44.5)
Smoking (%)		17.5	29.7	32.5
Antibiotic (%) (<3 month)		19	17	23.8
Previous infection with SARS-CoV-2 (%)		23	**5 * Ω**	19.5
Days since 2nd dose median (range) §		137 (53–289)	154 (35–311)	129.5 (36–242)

Demographics and risk factors of cohort 1 stratified into three groups according to first and second doses: homologous BNT162b2/BNT162b2 (BNT/BNT), homologous ChAdOx1 nCoV-19/ChAdOx1 nCoV-19 (ChAd)/ChAd), and heterologous BNT162b2/ChAdOx1 nCoV-19 (BNT/ChAd) vaccinations. Pairwise statistical comparisons between each group were made using a Kruskal–Wallis test with Dunn’s multiple comparisons correction for quantitative parameters and Fisher’s exact test for qualitative values. Bold indicates statistical significance (* *p* ≤ 0.05). § indicates time interval between sample collection and 2nd dose. Ω Indicates comparison with the BNT/BNT group.

**Table 2 vaccines-11-00744-t002:** Demographics of Cohort 2 Study Participants.

		Cohort 2 (*n* = 156)		
		BNT/BNT/BNT (*n* = 88)	ChAd/ChAd/BNT (*n* = 19)	BNT/ChAd/BNT (*n* = 49)
Parameters				
Gender (F:M)		34:54	7:12	23:26
Age (years) median (range)		**22 (18–38) * Φ**	23 (16–52)	**23 (17–49) * Ω**
BMI (kg/m^2^) median (range)		24.7 (16.7–40.9)	24.5 (15.8–36.3)	24.4 (16–34.5)
Smoking (%)		21.8	44.5	20.4
Antibiotic (%) (<3 month)		12.2	22.2	12.5
Previous infection with SARS-CoV-2 (%)		18.6	27.8	22.5
Days since 3rd dose median (range) §		81 (14–122)	75 (19–99)	72 (42–124)

Demographics and risk factors of cohort 2 who received a booster dose after first and second doses of homologous BNT162b2/BNT162b2 (BNT/BNT/BNT), homologous ChAdOx1 nCoV-19/ChAdOx1 nCoV-19 (ChAd/ChAd/BNT), or heterologous BNT162b2/ChAdOx1 nCoV-19 (BNT/ChAd/BNT) vaccination. Pairwise statistical comparisons between each group were made using a Kruskal–Wallis test with Dunn’s multiple comparisons correction for quantitative parameters and Fisher’s exact test for qualitative values. Bold indicates statistical significance (* *p* ≤ 0.05). § indicates time interval between sample collection and 3rd dose. Ω Indicates comparison with the BNT/BNT/BNT group. Φ indicates comparison with the BNT/ChAd/BNT group.

**Table 3 vaccines-11-00744-t003:** Multivariate Linear Regression for IgG Levels in Saliva.

Variables	ß	95% Cl	*p* Value
Previous infection with SARS-CoV-2 (yes)	0.065	0.048–0.082	**<0.0001**
Age	0.001	–0.001–0.002	0.2504
Gender (male)	0.005	–0.021–0.011	0.5257
BMI (kg/m^2^)	0.0003	–0.001–0.002	0.5892
Smoking (yes)	0.0062	–0.012–0.024	0.4995
Days since 2nd dose	–0.0001	–0.0002–0.0001	0.2947

The influence of potential risk factors on COVID-19-specific IgG levels in saliva of cohorts 1 and 2. ß = Standardized coefficient, indicates how much a dependent variable changes per unit variation of the independent variable. CI = Confidence interval. Bold indicates statistical significance.

## Data Availability

The data presented in this study are available on request from the corresponding author. The data are not publicly available due to privacy or ethical restrictions.

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
