# Peer review of "Salivary Antibody Responses to Two COVID-19 Vaccines following Different Vaccination Regimens"

_vaccines, 2023, doi:10.3390/vaccines11040744_

Round 1

Reviewer 1 Report

Comments on vaccines-2276392

Title: Mucosal Salivary Immune Responses to Different COVID-19 Vaccines and Vaccination Regiments

The present study investigated antibody responses to two COVID-19 vaccines (ChAdOx1 and BNT162b2) administered by different vaccination regimens in 301 saliva samples from vaccinated individuals in Saudi Arabia. The work is interesting and clinically relevant. But several major concerns should be addressed. 

Specific comments:

1.       The authors are suggested to check specific sIgA antibodies in the saliva samples to assess mucosal immune responses.

2.       More samples should be included in the cohorts to differentiate the salivary antibody responses among subgroups based on age and gender.

3.       The title should be change as “Salivary Antibody Responses to Two COVID-19 Vaccines Following Different Vaccination Regimens”.

4.       The manuscript should be revised by native English speaker.

Author Response

We would like to thank the reviewers for their positive comments and constructive advice that have helped us to improve the manuscript. Point-by-point replies to the specific comments raised are listed below.

Reviewer #1:

    The present study investigated antibody responses to two COVID-19 vaccines (ChAdOx1 and BNT162b2) administered by different vaccination regimens in 301 saliva samples from vaccinated individuals in Saudi Arabia. The work is interesting and clinically relevant. But several major concerns should be addressed.

Our reply: We thank the reviewer for the meticulous assessment of our study.

Specific comments:

  1. The authors are suggested to check specific sIgA antibodies in the saliva samples to assess mucosal immune responses.

Our reply:  Thanks for the suggestion. Yes, that would be very interesting to explore but since it’s been reported that the current COVID-19 systemic vaccine fails to induce specific IgA antibody responses in serum and were also found to be short-lived and detected only in a minority of subjects overtime in slaiva (Sterlin et al., Sci. Transl. Med. 10.1126/scitranslmed.abd2223, Alkharaan, H., et al.,  J. Infect. Dis., doi.org/10.1093/infdis/jiab256), we aimed here in this study to focus on salivary IgG mucosal response to Covid-19 vaccination.

  1. More samples should be included in the cohorts to differentiate the salivary antibody responses among subgroups based on age and gender.

Our reply:  Yes, unfortunately, recruitments for this study has been closed to accommodate other studies, but we performed several measures and tests to minimize the potential effect of age on salivary IgG by  targeting adult participants in the recruitment process and correlating also the variables of age and gender between all subgroups of cohorts 1 and 2 (tables 1 and 2), which showed that the majority of the group's participants were relatively at comparable age ranges and gender. Furthermore, age was analyzed as an independent factor in the multiple linear regression analysis and showed no correlation to IgG levels in saliva in our cohorts (table 3).

  1. The title should be change as “Salivary Antibody Responses to Two COVID-19 Vaccines Following Different Vaccination Regimens”.

Our reply: Thanks for the suggestion. We change it accordingly.

  1. The manuscript should be revised by native English speaker.

Our reply: Thanks, we have revised our manuscript and corrected the typos.

Reviewer 2 Report

The study presents an original work focusing on the efficacy of homologous or heterogous vaccine against SARSC-COV-2 through the detection of immunoglobulin IgG in the oral cavity in volunteers vaccinated with different protocols. In addition, the oral route may help to improve patient compliance, which is very important; particularly in situations such COVID-19 epidemics where a large part of the population needs to acquire immunity through vaccination

The study is very relevant in terms of design and methodology. Limits are evoked with suggested improvement in the next protocols.

Nevertheless, I have some issues related to the recruitment of the convalescent volunteers and the serums which were used for the validation of their IEA test.

Please find some suggestions to improve  your manuscript: 

- M&M section: 

1- please add some more details about the convalescent individual recruitement

2- Also for the 21 serums which were used for ELISA validation

Results section:

1- Please  remove sentences from 148 to 151 and replace in the discussion section

2- what is B in the Table 3

Discussion section

1- add refernce number to Mubarak et al., in line 186

2- add refernce number to Mubarak et al., in line 188

3- the same for Goel et al., line 194.

Author Response

The study presents an original work focusing on the efficacy of homologous or heterogous vaccine against SARSC-COV-2 through the detection of immunoglobulin IgG in the oral cavity in volunteers vaccinated with different protocols. In addition, the oral route may help to improve patient compliance, which is very important; particularly in situations such COVID-19 epidemics where a large part of the population needs to acquire immunity through vaccination

The study is very relevant in terms of design and methodology. Limits are evoked with suggested improvement in the next protocols.

Nevertheless, I have some issues related to the recruitment of the convalescent volunteers and the serums which were used for the validation of their IEA test.

Our reply: We thank the reviewer for the positive comments and constructive advice.

- M&M section:

  • please add some more details about the convalescent individual recruitement

Our reply: Thanks for the suggestion. We added the details. Please see lines 83-85.

  • Also for the 21 serums which were used for ELISA validation

Our reply: sorry if it was not sufficiently clear. As mentioned in lines 124-126.  we used the same saliva samples in both commercial and in-house assays to compare IgG-level responses.

Results section:

  • Please remove sentences from 148 to 151 and replace in the discussion section

Our reply: Thanks, we moved to lines 188-191 in the discussion section.

  • what is B in the Table 3

Our reply: Thanks, we describe it in the legend

Discussion section

  • add refernce number to Mubarak et al., in line 186

Our reply: it has been added, ref: 36.

2- add refernce number to Mubarak et al., in line 188

Our reply: it has been added, ref: 36.

3- the same for Goel et al., line 194.

Our reply: Thanks, done. ref: 16.

Reviewer 3 Report

The manuscript titled “Mucosal Salivary Immune Responses to Different COVID-19 2 Vaccines and Vaccination Regiments” is very well written. In this manuscript, Alkharaan et al., has collected 301 saliva samples from vaccinated individuals and arranged into 2 cohorts, depending upon the doses of vaccines received. They have observed that Salivary IgG antibody responses against different vaccines, whether homologous or heterogeneous vaccination regiments, showed similar levels in both cohorts. Boosting with BNT162b2 vaccine did not show an evident increase in mucosal IgG response whereby COVID-19 recovered subjects show higher salivary IgG than naïve, post-vaccination.

They have highlighted the importance for development of oral or intra-nasal vaccines to induce stronger mucosal immunity.

I recommend this article for the publication.

Author Response

The manuscript titled “Mucosal Salivary Immune Responses to Different COVID-19 2 Vaccines and Vaccination Regiments” is very well written. In this manuscript, Alkharaan et al., has collected 301 saliva samples from vaccinated individuals and arranged into 2 cohorts, depending upon the doses of vaccines received. They have observed that Salivary IgG antibody responses against different vaccines, whether homologous or heterogeneous vaccination regiments, showed similar levels in both cohorts. Boosting with BNT162b2 vaccine did not show an evident increase in mucosal IgG response whereby COVID-19 recovered subjects show higher salivary IgG than naïve, post-vaccination.

They have highlighted the importance for development of oral or intra-nasal vaccines to induce stronger mucosal immunity.

I recommend this article for the publication.

Our reply: Thanks for your review and nice comment

Round 2

Reviewer 1 Report

The revised manuscript can be accepted after polishing.